# Peripheral Iron Levels in Autism Spectrum Disorders vs. Other Neurodevelopmental Disorders: Preliminary Data

**DOI:** 10.3390/ijerph19074006

**Published:** 2022-03-28

**Authors:** Andrea De Giacomo, Silvia Medicamento, Chiara Pedaci, Donatella Giambersio, Orazio Valerio Giannico, Maria Giuseppina Petruzzelli, Marta Simone, Massimo Corsalini, Lucia Marzulli, Emilia Matera

**Affiliations:** 1School of Medicine, University of Bari “Aldo Moro”, 70124 Bari, Italy; silviamedicamento@gmail.com (S.M.); chiarapedaci@hotmail.it (C.P.); dona.215@hotmail.it (D.G.); maria.petruzzelli@uniba.it (M.G.P.); marta.simone@uniba.it (M.S.); massimo.corsalini@uniba.it (M.C.); lucia.marzulli@uniba.it (L.M.); emilia.matera@uniba.it (E.M.); 2Department of Prevention, Local Health Authority of Taranto, 74121 Taranto, Italy; oraziovaleriogiannico@gmail.com

**Keywords:** ASD, autism, iron status, ferritin, MCV

## Abstract

Autism Spectrum Disorder (ASD) is characterized by deficits in social skills and specific behaviors and interests. Among other environmental factors, iron may play a role in the development of ASD. The aim of this study is to compare the iron status of children with ASD with that of children affected by neurodevelopmental disorders other than ASD (OND). A total of 167 patients were enrolled, including 93 children with ASD and 74 children with OND. In the two groups, we determined ferritin, iron, transferrin, hemoglobin, HCT, and MCV in the serum. We found a significant difference in serum ferritin and MCV levels between the two groups (*p* < 0.05), with lower ferritin and higher MCV values in the ASD group. There was no significant association with the other variables. Our results may support the hypothesis of altered iron status in ASD, justifying more frequent examinations of blood iron parameters in these children.

## 1. Introduction

Autism spectrum disorder (ASD) is a neurodevelopmental disorder characterized by deficits in communication and social interaction. There are also restricted, repetitive patterns of behavior, interests, or activities. The expression of these impairments is variable in scope and severity and often changes with the acquisition of other developmental skills. Autistic symptoms appear in childhood and negatively affect social, occupational, or other domains [1,2]. Nowadays, one in every 44 newborns is diagnosed with ASD, and the male-to-female ratio is four to one [3]. The etiopathogenesis of ASD is still unknown, but there is much evidence for the role of epigenetic, neurobiological, genetic, neurological, and hormonal factors [4]. There are many hypotheses about the role of genetic polymorphisms and abnormalities in specific genes, neurotransmitter dysfunction, brain connectivity abnormalities, and prenatal infections or toxin exposure [5]. In addition, there is increasing evidence of the possible role of environmental (non-genetic) factors in the etiology of ASD, such as prenatal factors (gestational diabetes, maternal bleeding and medications during pregnancy, intrauterine infections) and perinatal and neonatal factors (fetal presentation, umbilical cord complications, fetal distress, perinatal cyanosis, twin pregnancy, maternal internal bleeding, low birth weight, low Apgar score) [6]. Closely related to such factors, metabolic disorders may contribute to the occurrence of ASD [7]. For this reason, iron deficiency (ID), which is the most prevalent nutrient deficiency according to the World Health Organization and affects approximately two billion people worldwide [8], is currently under investigation as a possible etiopathogenetic factor of ASD. Iron is an essential component of cellular metabolism, DNA replication and repair, and the development of various organs and systems, including the central nervous system. The diagnosis of ID in children is based on the following: decreased serum iron [<60 micrograms/dL], hemoglobin (Hb) [<11.3 g/dL], and serum ferritin [<20 micrograms/dL] [9].

Some studies reported an association between ASD and ID [10,11,12,13,14], but literature data on this topic are still limited. Several authors reported that low maternal iron levels are associated with a high risk of autism [15]. ID can be both a concomitant cause of this neurodevelopmental disorder and a consequence of one of the typical symptoms of ASD, hypersensitivity, which often affects the intraoral sense of taste. The diet of children with autism is often characterized by food selectivity. Nutritional difficulties can affect the child’s health status [16], as severe malnutrition in children with ASD can lead to stunted growth [17]. Restricted nutrition, in turn, can lead to decreased iron absorption and the development of ID, which is more common in preschool children with autism, as restricted food choices are more prevalent in this group of children [10].

The aim of our case-control study is to investigate the iron status of children with ASD compared to children with other neurodevelopmental disorders, determining the following biochemical and hematological indices used to diagnose and differentiate anemia: serum iron, ferritin, transferrin, Hb, mean corpuscular volume (MCV), and hematocrit (HCT).

## 2. Materials and Methods

### 2.1. Participants

We retrospectively analyzed the records of 167 patients referred to the Child Neuropsychiatry Unit of the University of Bari “Aldo Moro” from 2018 to 2021.

The enrolled patients were divided into the following two groups: the ASD group, which included 93 children diagnosed with ASD, and the OND group, which included 74 children diagnosed with a neurodevelopmental disorder other than ASD.

All patients met the following inclusion criteria: age up to 18 years, both sexes, diagnosis of a neurodevelopmental disorder [intellectual disability, ASD, movement disorder, attention deficit hyperactivity disorder (ADHD), specific learning disorder (SLD), language disorder]. The exclusion criteria for both groups were the following: genetic syndrome, neurological pathologies, brain malformations, and conditions that may cause iron deficiency or iron overload (bleeding, restricted or selective diet, use of proton pump inhibitors, H2 antihistamines and antacids, previous duodenal surgery, Crohn’s disease, celiac disease, inflammatory conditions, cancer, transfusions, hemochromatosis). In addition, patients who were breastfeeding, using oral contraceptives, or pregnant were excluded from the study.

The diagnosis of ASD and other neurodevelopmental disorders was made by child and adolescent psychiatrists according to the criteria of the *Diagnostic and Statistical Manual of Mental Disorders* (DSM-5). The diagnostic process was based on observation of the child, clinical interview with the parents, and the results of the following standardized and validated tests in Italian:Leiter International Performance Scale—Third Edition (Leiter 3) [18],Wechsler Intelligence Scale for Children—Fourth edition (WISC IV) [19],Wechsler Preschool and Primary Scale of Intelligence—III (WPPSI III) [20],Child Behavior Checklist (CBCL) [21],Conners’ Rating Scales-Revised (CRSR) [22],SNAP-IV Rating Scale [23],Test TVL—(Language Valuation Test) [24],MT-3 Clinical tests for the assessment of reading and comprehension skills for primary and lower secondary school [25],AC MT test for 6–11 years [26],DDE-2–Battery for the evaluation of Dyslexia and Disortography in evolutive age [27],Vineland Adaptive Behavior Scales (VABS) [28],Autism Diagnostic Interview-Revised (ADI-R) [29], andAutism Diagnostic Observation Schedule, Second Edition (ADOS-2) [30].

### 2.2. Procedures

Clinical examinations included the following: anamnestic interview, physical and neurological examinations, and laboratory blood tests on all patients.

Neurologic examination of all participants revealed no evidence of focal neurologic symptoms.

Laboratory blood tests included biochemical (serum ferritin, iron, transferrin, Hb) and hematological tests (HCT, and MCV) to assess iron metabolism. Most of the iron that enters plasma comes from red blood cell destruction, while the remainder comes from reserve iron and only a small amount from dietary iron. Both intestinally absorbed and deposit-derived iron are transported in plasma in association with transferrin, a protein responsible for transporting iron to the bone marrow for incorporation into the Hb. Ferritin is the main iron storage protein in cells. Its concentration in the blood therefore reflects the extent of mineral reserves in the body. HCT expresses the ratio between the liquid part of the blood and the corpuscular part, that is the part occupied mainly by red blood cells. This index indicates the average volume of red blood cells.

Venous blood was drawn from all patients in the morning between 8:00 and 9:00 a.m., after they had fasted overnight. To avoid exposure to the prick, all patients were drawn in the supine position. Lidocaine cream was applied 20 min before blood collection.

We defined ferritin deficiency, iron blood deficiency, Hb deficiency, HCT deficiency, and MCV deficiency as values below the reference range established by age and sex. Transferrin deficiency was determined by transferrin levels below 200 mg/dL.

Written informed consent was obtained from parents, and the study was approved by the local ethics committee of the University Hospital of Bari.

### 2.3. Statistical Analysis

Statistical analysis was performed using R 4.0.2 (released 22 June 2020). Statistical significance was set with alpha at 0.05. Numeric variables were expressed as median and IQR and compared with the Wilcoxon Rank Sum test to account for non-normality, which was assessed with the Shapiro Wilk test.

Categorical variables were expressed as absolute and relative frequencies (%) and compared with the Chi-Square test.

The 6 independent variables (ferritin, iron, transferrin, Hb, HCT, MCV) were categorized according to the median value of the included patients and, to analyze the association between them and the presence of ASD, 6 logistic regression models with estimation of odds ratios (OR) were fitted. Another multivariable logistic regression model was then fitted with the combination of variables with a *p* value < of 0.1 in the logistic models. All models were adjusted for age and sex.

## 3. Results

The ASD group included 93 patients, whereas 74 subjects with other neuropsychiatric disorders were included in the control group. The OND group included children diagnosed with a neurodevelopmental disorder other than ASD (17 with intellectual disability, 25 with ADHD, 19 with SLD, 10 with language disorder, 2 with tic disorder, and 1 with Tourette syndrome).

The total sample consisted of 117 male patients (70%) and 50 female patients (30%).

Baseline data of the participating patients are shown in Table 1. Age, ferritin and Hb were significantly higher in the control group (*p* < 0.05).

Multiple logistic regression results are shown in Table 2 and Table 3. Taking age and sex into account, ferritin > 24 ng/mL was negatively associated with the ASD group (*p* = 0.0322). The model with ferritin > 24 ng/mL and Hb > 13 g/dL, adjusted for age and sex, showed no significant association for Hb > 13 g/dL, which was excluded. When we fitted the multiple models with ferritin > 24 ng/mL and MCV only, adjusted for age and sex, both variables showed a statistically significant association with ASD (ferritin > 24 ng/mL: OR = 0.46 (0.23–0.89), *p* = 0.0230; MCV > 80 fL: OR = 2.13 (1.05–4.50), *p* = 0.0417).

## 4. Discussion

In the present study, we examined iron status in a sample of children with ASD compared with a sample of children with other neurodevelopmental disorders. The first important finding was the statistically significant difference in serum ferritin levels between ASD patients and the OND group, with lower ferritin levels in the ASD group. More specifically, ASD patients were more likely to have hypoferritinemia than children with other neurodevelopmental disorders in over 80% of cases. The second important finding was the statistically significant difference in MCV levels between ASD patients and the OND group, with higher MCV levels in the ASD group. There were no statistically significant differences between the median values of the other biomarkers of iron status, although serum Hb levels and hematocrit were slightly lower and iron and transferrin levels were higher in ASD patients than in the control group.

Iron is an essential component of several proteins and enzymes that perform important functions in cell metabolism and survival, including protecting the genome from damage and mutations. It also plays a role in the development and activities of various organs and systems. In particular, in the brain, iron is involved in the myelination of white matter and in the function of various neurotransmitter systems, such as dopamine, norepinephrine, and serotonin. For these reasons, iron status is thought to be likely related to the occurrence of various neuropsychiatric disorders, including autism [8].

It is important to remember that there is no test that is sufficient in itself to evaluate iron status with or without anemia. In fact, any test that assesses iron status reflects changes in different compartments of endogenous iron (storage, transport, and metabolic-functional), is influenced by different levels of iron deficiency, and has an overlap between normal and pathological values. Scientifically based data exist only for ferritin, which functions as an iron storage protein, in terms of the relationship between its serum concentration and marrow iron content, providing a reliable indication of martial heritage. Its decrease, in the absence of hyposideremia and normal transferrin levels, indicates initial depletion of iron stores [31].

In our study, we found lower serum ferritin levels in ASD patients and hypoferritinemia in more than 50% of autistic children, in accordance with the results of other authors [32,33]. We have not studied the correlation between serum ferritin levels and the severity of autistic symptoms, but it is possible that ferritin could influence the clinical expression of autism and various neurological and psychiatric disorders. Several authors pointed out that lower ferritin levels (<20 ng(mL) are found in patients with severe ASD, while other studies found no correlation between serum ferritin levels and autistic symptom scores [10]. Conflicting results are also seen regarding the relationship between ferritin and behavioral manifestations of other neurodevelopmental disorders such as ADHD (which were part of our OND group) [34].

Our ASD group had significantly higher MCV values compared with the OND group. A medium-high MCV value indicates the presence of red blood cells that are too large compared to the norm (macrocytes). This may be due to several causes, including a deficiency of vitamin B12 and folic acid (essential for the correct synthesis of red blood cells). Vitamin B12 is an essential component for DNA synthesis and for the production of cellular energy. Deficiency of vitamin B12 causes gastrointestinal, hematologic, neurologic, and psychiatric symptoms such as motor dysfunction, sensory and memory deficits, and cognitive impairment. Although evaluation of vitamin B12 and folate was not our main purpose, such data may support the hypothesis of Adams JB et al. [7], which compared the nutritional and metabolic status of children with ASD with that of neurotypical children. The results of this study showed that ASD patients had many alterations in their nutritional and metabolic parameters (vitamins, minerals, plasma amino acids, plasma glutathione, biomarkers of oxidative stress, methylation, sulfation, and energy production) that could also be associated with variations in the severity and etiology of autism. The underlying causes of this association are not yet clear.

In our study, Hb levels and HCT in ASD patients were slightly lower than in the control group, while serum levels of iron and transferrin were higher than in patients with other neurodevelopmental disorders, although the difference was not statistically significant. No patients were diagnosed with anemia. Even if we refer to the presence of ID and IDA, the previous literature provides rather conflicting results.

The prevalence of ID in children with ASD, which has been estimated to be 7–52% in South Wales, Turkey, and Canada, is higher than in children with normal development, especially in patients with severe ASD or with ASD and cognitive impairment [11]. A meta-analysis conducted by Tseng et al. showed that there was no specific difference in iron status between children with ASD and the general population, although children with ID were more likely to be diagnosed with ASD than those without ID [5], suggesting a possible interaction between ASD and IDA. Thus, inadequate dietary habits in these subjects could warrant IDA, and the food selectivity of many children with ASD could contribute to IDA, especially at a young age [14].

## 5. Conclusions

This study supports the hypothesis of altered iron status or initial increased iron requirements without anemia in ASD children and highlights the need to monitor blood iron parameters in these patients in clinical practice.

The limitations of this preliminary study are the small sample size, the lack of another healthy and age-matched control group (although there is a lack of studies comparing different neurodevelopmental disorders), and its retrospective nature.

Determination of iron and, more generally, metabolic status is complex because it depends not only on intake but also on digestion, absorption, metabolic processing, and requirements. However, future observational prospective studies should be conducted with larger samples, healthy controls, markers of iron absorption, and dietary history to allow generalizability of results and to better elucidate the nature of the association between ASD and iron metabolism (random, environmental, or genetic). Iron supplementation/treatment of children with ASD is not recommended without documenting iron status parameters and monitoring laboratory exams because it could lead to side effects such as constipation, gastrointestinal distress, and cardiological problems. Future studies should evaluate if and when treatment/supplementation with iron might be useful in autistic patients.

## Figures and Tables

**Table 1 ijerph-19-04006-t001:** Baseline data of patients, Wilcoxon Rank Sum test and Chi Square test.

N = 167	OND (*n* = 74)	ASD (*n* = 93)	*p*
	Median (IQR) or *n* (%)	Median (IQR) or *n* (%)	
Male sex	46 (62.2%)	71 (76.3%)	0.0691
Age (years)	11.0 (6.0)	7.0 (5.0)	0.0001
Ferritin (ng/mL)	28.5 (21.0)	22.0 (18.0)	0.0249
Iron (ug/dL)	71.0 (34.0)	75.0 (51.0)	0.3345
Transferrin (mg/dL)	278.5 (60.0)	286.0 (60.0)	0.2540
Hb (g/dL)	13.4 (1.3)	12.9 (1.2)	0.0361
MCV (fL)	79.8 (4.2)	79.8 (6.9)	0.7339
HCT (%)	39.4 (4.5)	38.1 (3.6)	0.0687

**Table 2 ijerph-19-04006-t002:** Results of multiple logistic regression for hematologic variables and presence of ASD, adjusted for age and sex.

N = 167	OR (95%CI)	*p*
Ferritin > 24 ng/mL	0.49 (0.25–0.94)	0.0322
Iron > 73 ug/dL	1.20 (0.62–2.32)	0.5887
Transferrin > 285 mg/dL	1.37 (0.72–2.66)	0.3440
Hb > 13 g/dL	0.56 (0.28–1.12)	0.1008
MCV > 80 fL	1.98 (0.99–4.07)	0.0581
HCT > 39%	0.63 (0.32–1.26)	0.1942

**Table 3 ijerph-19-04006-t003:** Results of multiple logistic regression for hematologic variables and the presence of ASD, reciprocally adjusted and adjusted for age and sex.

N = 167	OR (95%CI)	*p*
Ferritin > 24 ng/mL	0.46 (0.23–0.89)	0.0230
MCV > 80 fL	2.13 (1.05–4.50)	0.0417

## Data Availability

Not applicable.

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
