# Peer review of "Peripheral Iron Levels in Autism Spectrum Disorders vs. Other Neurodevelopmental Disorders: Preliminary Data"

_ijerph, 2022, doi:10.3390/ijerph19074006_

Round 1

Reviewer 1 Report

All concerns were cleared.

Author Response

Thank you for reviewing the article

Reviewer 2 Report

All concerns were cleared.

Author Response

Thank you for reviewing the article

Reviewer 3 Report

Brief summary

The authors are investigating iron levels and associated blood parameters in children with Autism Spectrum Disordes (ASD) and other neurodevelopmental disorders. They found lower levels of serum ferritin levels ins children with ASD compared to children with other neurodevelopmental disorders. For the other measured blood parameters no significant differences were found.

General comments

Overall, this is an interesting and well written article. However, in my opinion a detailed description of all the tests that have been performed is not necessary in the methods section (Line 89-158), since those are not mentioned in the results or discussion section. If no results for those test are given, I think a list with the names of the performed tests should suffice. Currently you are describing 13 tests in detail (which comes to more than 1 page) which are not relevant for the content of the paper. Again, in my opinion listing the tests with name and reference, without further description would be more than sufficient and more reasonable in light of the content of the paper.

Specific comments

Line 46: it should say "affects about 2 billion people worldwide" - delete the "of" and add a "worldwide".

Line 50/51: add an "and" before "serum ferritin".

Line 59: There is a space too much before "iron intake"

Line 88: it should say "and validated tests in italian:"

Line 89-158: Please shorten. Just list the names of the tests with the respective references and leave any further description out.

Line 165: There is a space too much before "Most".

Line 222-224: Please check that sentence again. Suggestion for revising: "The first most important finding was the statistically significant difference in serum ferritin levels between ASD patients and the control group; with lower ferritin levels in the ASD group."

Line 225-227: Please check that sentence again. Suggestion for revising: " The second important finding was the statistically significant differerence in MCV between ASD patients and the control group; with higher MCV levels in the ASD group."

Line 255: "we founf lower ferritin levels" --> "lower" should be placed before "serum ferritin".

Line 261-263: Suggestion for revising: "Seveal authors pointed out that lower ferritin levels (<20 ng(mL) are found in patients with more severe ASD, whereas other studies did not find a correlation between serum ferritin levels and autistic symtom scores."

Line 267: I would suggest to use "group" instead of "sample". Add a "the" before "control group".

Line 273: Add an "and" before "cognitive impairment".

Line 276: "Results of that study..."

Line 277: I would suggest to write "metabolic parameters" instead of "metabolic status".

Line 280: End the sentence after "...and etiology of autism." And then start a new sentence "The underlying causes of this relationship have not yet been clarified."

Line 288: Please check that sentence again. It is not clear. Should it say "...in typically developing children AND in patients with severe ASD..." - The way the sentence is written right now, it does not make sense. Please revise.

Line 304: I would suggest to write "However" instead of "Although that".

Line 309: Iw ould suggest to write "monitoring" instead of "a monitor of".

Line 310: Please add an "and" before cardiological problems.

Author Response

Dear reviewer,

thank you for your comments. As you requested, we listed the tests used for the evaluation of the enrolled patients with name and reference, without further description. In addition, we changed the sentences from line 46 to 310 as you suggested. We are sure that your considerations will ameliorate the form and quality of our manuscript.

Kind regards,

Andrea De Giacomo

Reviewer 4 Report

The present study entitled “Peripheral Iron Levels in Autism Spectrum Disorders vs Other 2 Neurodevelopmental Disorders: Preliminary Data” by Giacomo et al., presenting very interesting finding about ASD and found a topic interesting for general audience. They have addressed serum ferritin, serum iron, transferrin, and hemoglobin in the peripheral blood of ASD and control groups and found a significant difference in serum levels of ferritin between the two groups (p<0.05) with lower levels in the ASD group. However, there are very minor comments.

  1. Carefully check English throughout MS.
  2. Make concise abstract and highlight the findings of MS.
  3. Authors have included patients who suffered with other neurodevelopmental disorders other than ASD as a control. Why authors didn’t include normal healthy subjects of similar ages to make better interpretation?
  4. Why authors have written control patients if they have other disorders?
  5. Discussion could be shorten and focused on their own findings and relevance.

Author Response

Dear reviewer,

thank you for the useful comments which certainly improved the quality of this manuscript.

1-2. As you suggested, we checked English throughout MS, maked concise abstract and highlighted the findings of MS.

  1. We did not include normal healthy subjects in our study because in our clinic we only visit subjects with psychopathological, neurological and neurodevelopmental pathologies. Moreover, it is very difficult for healthy subjects to agree to undergo a blood sample.

  1. As you have rightly pointed out, the sample of subjects with other neurodevelopmental disorders was named OND.
  2. We tried to make discussion shorten and focused on our findinggs and their relevance.

Kind regards,

Andrea De Giacomo

This manuscript is a resubmission of an earlier submission. The following is a list of the peer review reports and author responses from that submission.

Round 1

Reviewer 1 Report

The authors report a slight elevation of blood transferrin in ASD patients compared to a control group composed by subject diagnosed with other neuropsychiatric conditions. Only transferrin is significantly different (increased) in the ASD cohort compared to the control group, while blood iron is normal. Based on this, the authors speculate that reduced transferrin may contribute to ASD pathogenesis and be exploited in therapeutic perspective.

The study has several major limitations:

1- lack of a true control group including age-matched normal subject.

2- the presence of a substantial subgroup (nearly 50%) of subjects in the ASD cohort with selective diet which may drive statistical difference in transferrin levels (although iron is normal).

3- Selective diet is not detailed or used as a variable: is it a diet leading to reduced transferrin synthesis by the liver? ASD cohort has specific transferrin deficiency, while iron is normal. The effect of diet on the transferrin level must be controlled. If it is simply a matter of diet iron supplementation, speculations on the association between iron status and ASD development are not supported by evidence, let alone the suggestion of iron supplements as an ASD therapy.

4- Iron blood levels are normal in ASD subjects, suggesting that there is no dietary deficiency. Therefore, reduced transferrin could have genetic basis, and this hypothesis would be worth testing, but in the manuscript there is no discussion of this.

4- Although transferrin levels, following multiple regression analysis is significantly increased in the ASD cohort,  the OR is 1, which suggests no association with the condition.

5- The study has serious conceptual weakness: ASD is a neurodevelopment disorder. The hypothesis that environmental factors (such as the availability of key nutrients) drives disease has a strong rationale, but only during pregnancy or early post-natal age. The pathophysiological involvement of dietary deficits (i.e., iron) at the tested ages (7-11 years), when disease is manifest, is not supported by evidence. As a consequence, the rationale for a therapeutic exploitation of such factors is weak.

Author Response

1st Revisor

The authors report a slight elevation of blood transferrin in ASD patients compared to a control group composed by subject diagnosed with other neuropsychiatric conditions. Only transferrin is significantly different (increased) in the ASD cohort compared to the control group, while blood iron is normal. Based on this, the authors speculate that reduced transferrin may contribute to ASD pathogenesis and be exploited in therapeutic perspective.

The study has several major limitations:

1- lack of a true control group including age-matched normal subject.

  • Certainly, as you have rightly pointed out, the lack of a further healthy control group is a limitation of our study, as indicated in our discussion.

2- the presence of a substantial subgroup (nearly 50%) of subjects in the ASD cohort with selective diet which may drive statistical difference in transferrin levels (although iron is normal).

  • We decided to delete the sentence relating to selective diet in the results section, because the medical records did not report any information relating to the characteristics of the patient's diet for most of our study sample. When reported, information on diet was often limited to describing the presence of a "varied and regular diet", or a "selective diet", without indicating which foods were present / absent in the child's diet in most of cases. For these reasons, it was not possible to consider diet as a confounding factor in our statistical analysis, nor to draw conclusions on the possible role it played in influencing the iron status of the child. Anyway, we intend to carry out a prospective study conducted with a rigorous methodology of collecting clinical data to investigate the role of nutrition on alterations of the iron status in children with ASD. We have included the following sentence in the limitations: “The limitations of this work are the small sample size,  the  lack of a further healthy and age matched control group, and the lack of dietary assessment”.

3- Selective diet is not detailed or used as a variable: is it a diet leading to reduced transferrin synthesis by the liver? ASD cohort has specific transferrin deficiency, while iron is normal. The effect of diet on the transferrin level must be controlled. If it is simply a matter of diet iron supplementation, speculations on the association between iron status and ASD development are not supported by evidence, let alone the suggestion of iron supplements as an ASD therapy.

  • Please see the answer given above.

4- Iron blood levels are normal in ASD subjects, suggesting that there is no dietary deficiency. Therefore, reduced transferrin could have genetic basis, and this hypothesis would be worth testing, but in the manuscript there is no discussion of this.

  • Thanks to your comment, we enriched our discussion about the association between increased transferrin and ASD condition, underlying that multiple interpretations are possible.

4- Although transferrin levels, following multiple regression analysis is significantly increased in the ASD cohort,  the OR is 1, which suggests no association with the condition.

  • Accordingly with your appropriate consideration, we have re-written the discussion considering that the risk of having higher transferrin level is only 0.01 fold increased in ASD children compared to our non-ASD control group, pointing out a slightly higher correlation between the conditions. However, among limitations and future directions, we elucidated that a larger sample study could be useful to further validate our results.

5- The study has serious conceptual weakness: ASD is a neurodevelopment disorder. The hypothesis that environmental factors (such as the availability of key nutrients) drives disease has a strong rationale, but only during pregnancy or early post-natal age. The pathophysiological involvement of dietary deficits (i.e., iron) at the tested ages (7-11 years), when disease is manifest, is not supported by evidence. As a consequence, the rationale for a therapeutic exploitation of such factors is weak.

  • Effectively, we did not intend to suppose that iron dietary deficiency in the child could cause pathophysiological alterations in autism. For this reason and in order to respond to all reviewers’ requests, we have re-written our discussion to better clarify our hypothesis and the potential meaning of our results.

Reviewer 2 Report

Comments:
The topic of the possible relationship of iron status with ASD is
interesting.
However, the manuscript lacks robustness.
- The weakest is the Results section:
- Although they have preliminary data, these can be exploited further.
- The presentation of the tables is very limited; it presents means or
medians, but also, for example, the number and percentage of children
with low, low transferrin, low hemoglobin, etc. could be presented.
- In table 1, there is no footnote, it is not clear which are n (%) and
which are medians and IQR. The statistical test used does not put in the
title.
- Table 2 is incomplete. It does not have proper column headings, it
doesn’t have footnotes, or the adjustment variables that were used.
- It needs English edition 

Author Response

2nd Revisor

The topic of the possible relationship of iron status with ASD is interesting.

However, the manuscript lacks robustness.

- The weakest is the Results section:

- Although they have preliminary data, these can be exploited further.

  • Accordingly with your appropriate consideration, we clarify that our results are only preliminary because of small sample size and lack of some important clinical information, such as dietary patterns. We have highlighted this aspect in our discussion, among limitations and future directions.

- The presentation of the tables is very limited; it presents means or medians, but also, for example, the number and percentage of children with low, low transferrin, low hemoglobin, etc. could be presented.

  • According to your suggestion, we have enriched table 1 with the percentage of children with biochemical values under the normal range, pointing out a significant difference in hypoferritinemia prevalence.

- In table 1, there is no footnote, it is not clear which are n (%) and which are medians and IQR. The statistical test used does not put in the title.

  • We have modified tables according to your indications.

- Table 2 is incomplete. It does not have proper column headings, it doesn’t have footnotes, or the adjustment variables that were used.

  • We apologize for the misunderstanding, but in our version of the manuscript there are column headings. We added the adjustment variables in the footnotes.

- It needs English edition

  • We have edited English language with the help of an expert.

Reviewer 3 Report

The paper should be revised. Please correct followings before publication.

LINE 41
please add the updates of recent ASD prevalance using CDC information

Line 43
the authors described "The etiopathogenesis of ASD is still unknown, but 
42 many evidence have shown the role of epigenetic, neurobiological, genetic, neurological and hormonal factors. 
43 [4]"
is ref [4] relevant?
I think it is not relevant. please add the relevant ref. 

In Table 1
Authors performed wilcoxon rank sum test.
is it correct? please check the data normality. 
I think the nubmer of each group (n=68 and n=52) is enough to meet data normality.

line 154
Previous literature about the association between ASD and iron levels ..
please clarify which "Previous literature"

Line 177
other neuropsychiatric disorders : please clarify the "which neuropsychiatric disorders"

According to the study, no sinigifant difference between groups were observed except Transferrin in table 2.
however, most of part in discussion section was for the issue of iron (line 181- 187)
please discuss more regarding the role of transferrin.

Author Response

3rd Revisor

The paper should be revised. Please correct followings before publication.

LINE 41
please add the updates of recent ASD prevalence using CDC information

  • Thanks to your suggestion, we added the most recent ASD prevalence according to CDC (reference number 3).

Line 43
the authors described "The etiopathogenesis of ASD is still unknown, but many evidence have shown the role of epigenetic, neurobiological, genetic, neurological and hormonal factors. 
is ref [4] relevant?
I think it is not relevant. please add the relevant ref. 

  • The ref [4] was not relevant, we added the right reference ([4]: Shreeya, G; Bichitra, NP. Autism spectrum disorder: Trends in research exploring etiopathogenesis. Psychiatry and Clinical Neurosciences, 2019; 73: 466-475).

In Table 1
Authors performed wilcoxon rank sum test.
is it correct? please check the data normality. 
I think the nubmer of each group (n=68 and n=52) is enough to meet data normality.

  • Unfortunately, as described in methods section, we conducted Shapiro-Wilk test that demonstrated the non-normality of distribution of the evaluated parameters in our study samples. Anyway, we elucidated in our discussion that larger sample will be recruited in future studies to further validate our results.

line 154
Previous literature about the association between ASD and iron levels ..
please clarify which "Previous literature"

  • We modified our discussion and reported the previous literature about ASD and ID in introduction, with the appropriate references in brackets.

Line 177
other neuropsychiatric disorders : please clarify the "which neuropsychiatric disorders"

  • Thanks to your request, we clarified which are the other neurodevelopmental disorders in the results section.

According to the study, no sinigifant difference between groups were observed except Transferrin in table 2. However, most of part in discussion section was for the issue of iron (line 181- 187). Please discuss more regarding the role of transferrin.

Thank you for your notification. In order to make the discussion more consistent with our results, we changed it and we added some considerations about the possible meaning of transferrin alterations in patients suffering from autism spectrum disorder.

Round 2

Reviewer 1 Report

The manuscript received minor revisions compared to the original submission, mainly in the form of attenuation of conclusions that this reviewer found not supported by evidence. The discussion was entirely re-written base on these changes. However, the original weakness of the manuscript has not been addressed: sample size is too small, poorly characterized population, a healthy control group is lacking. A small difference in a single blood parameter (transferrin) between the groups is really not sufficient for a journal publication. I suggest the authors build on this interesting (but very preliminary) observation by setting up a prospective study where they can obtain a better characterization of the population under analysis and support more solid conclusions. Once they have confirmed their preliminary finding, they should answer the question: is this a genetic trait or merely the consequence of a restricted diet? The first scenario would be very relevant, the second much less but still useful for clinical purposes. In the absence of sufficient evidence to support either hypothesis, the manuscript does not meet the minimal requirements for a peer-reviewed publication in an international journal.

Reviewer 2 Report

The manuscript improved. The results and discussion are better presented.

Reviewer 3 Report

All concerns were cleared.

It was worthy of publication.